# Mussel-Inspired Adhesive and Self-Healing Hydrogel as an Injectable Wound Dressing

**DOI:** 10.3390/polym14163346

**Published:** 2022-08-17

**Authors:** Kai-Yi Chang, Ying-Nien Chou, Wei-Yu Chen, Chuh-Yean Chen, Hong-Ru Lin

**Affiliations:** Department of Chemical and Materials Engineering, Southern Taiwan University of Science and Technology, Tainan 71005, Taiwan

**Keywords:** hydrogel, N-isopropyl acrylamide, dopamine, mussel-inspired, injectable hydrogel, hydrophobic association, self-healing, adhesion

## Abstract

This study develops a multi-functional hydrogel with a dual injection system based on the adhesive and self-healing properties of the byssus excretion found in mussels. Through precisely controlling the composite cross-linking hydrophobic association (HA) structure composed of A and B solutions, a high-strength, temperature-sensitive injectable hydrogel can be obtained, and it has good self-healing properties. The main composition of A solution contains the surfactant SDS, which can form amphiphilic micelles, the strength increasing component stearyl methacrylate (C18), and NIPAAm, which provides thermo-sensitivity. Solution B contains dopamine acrylate (DAA), which has self-healing properties, and ferric chloride (FeCl_3_), which is a connecting agent. The rheological behavior shows that when the temperature is increased from 25 °C to 32 °C, the gel can be completed in seven minutes to form a composite hydrogel of NIPAAm-DAA-HA. When NMR identification was conducted on composite DAA, it was found that when comparing DAA and dopamine hydrochloride there were new peaks with specific characteristics, which confirm that this study successfully prepared DAA; swelling tests found that swelling could surpass a rate of 100%, and a higher ratio of crosslinking agent decreased the amount of moisture absorbed; the results of the compression test showed that the addition of hydrophobic micelles C18 effectively enhanced the mechanical properties of hydrogel, allowing it to withstand increased external stress; the adhesiveness results show that an increase in the catechol-Fe^3+^ concentration of the NIPAAm-DAA-HA hydrogel results in an increased adhesiveness of 0.0081 kg/cm^2^ on pig skin; the self-healing tests show that after taking damage, NIPAAm-DAA-HA hydrogel can be reactivated with catechol-Fe^3+^ and self-heal at a rate of up to 70% after 24 h; antibacterial tests show that hydrogel has good bacterial resistance to against *E. coli*, staphylococcus epidermidis, and bacillus cereus; through in vitro transdermal absorption, it can be seen that the release ability of drugs within the hydrogel can reach up to 8.87 μg/cm^2^. The NIPAAm-DAA-HA hydrogel prepared by this study performed excellently in both adhesion and self-healing tests. The thermo-sensitive and antibacterial properties can be applied to the treatment of deep wounds and address some of the flaws of traditional wound dressings.

## 1. Introduction

Skin is the human body’s largest and most important organ. It protects against external microbes and bacteria that attempt to invade the body, prevents the body from dehydration, and maintains the nutrients needed by the human body. Damaged and defective skin will lose its protective qualities, allowing microbes to invade injuries, cause infections, obstruct the healing process of injuries, and impact daily life [1,2]. Skin has exceptional self-repair ability when lightly injured, but the occurrence of severe defects or diabetes, malnutrition, and specific diseases will greatly impair its ability to repair tissue. This is especially apparent in deep wounds that require long periods to heal, eventually turning into chronic wounds that become even harder to heal [3]. 

The general concept of treating injuries is keeping wounds in a dry environment, but a study by Dr. George Winter proved that wounds kept in moist environments healed faster than those in dry environments, and the healed wound resulted in reduced scarring [4]. Wound dressings should have the effects of air flow, moisturization, and sterilization, keeping the surface of wounds moist to prevent dryness and infection to speed up the healing rate of wounds [5]. Current research on wound dressings has made great developments, but their healing effects are still not comparable to that of skin cells [6]. In the initial stage of wound formation, the increased generation of exudate means that using traditional dressings such as bandages, gauze, or cotton will continuously absorb liquids, and the removal of the dressing often causes secondary injury due to adhesion to the wound’s surface. In order to overcome the limitations of traditional dressings, subsequent research has shifted towards developing wound dressings that can maintain a moist environment and absorb tissue exudate.

Wounds that are relatively shallow can be treated with topical medication. However, the self-healing ability of skin is limited, and while skin does have regenerative properties they are insufficient for healing deep wounds completely. Autologous skin grafts are often used to treat deep wounds, but this is determined by the availability of the donor site and there are some surgical risks. Furthermore, post-surgical scars may impact mobility, and uneven skin structure after transplantation may result in weaker skin regeneration, which is one of the clinical issues faced by those who suffer acute and chronic wounds [7]. Without question, an injectable hydrogel that has low invasiveness and can fill irregular shapes would satisfy the requirements for quickly closing wounds, promoting healing, and reducing the formation of scars. Hydrogel is also equipped with a drug carrying capacity and can be used to achieve drug injection, having shown some extraordinary properties in biomedical applications. Even so, further development of injectable hydrogel in the field of biomedicine cannot ignore some basic requirements in addition to developing new properties and advanced functionality. First, injectable hydrogel must accurately generate a sol-gel transition at the target site, and cannot gel too slowly before displaying viscosity. However, a sol-gel transition that occurs too quickly during the injection process may cause the fluid to block the needle used for the injection. On the other hand, injecting fluids with high viscosity requires high-pressure administration, which may cause hand fatigue for doctors and discomfort for patients [8]. In order to improve the issue of sol-gel transitions blocking needles during the injection process, we changed the single syringe to a double syringe that injects two solutions, which mix and quickly form a hydrogel when reacting to body temperature. The hydrogel is given properties of adhesion and self-healing to ensure it can completely cover a wound and repair hydrogel that is damaged due to activity.

There are now many studies of wound treatment using injectable hydrogels. Y-F Tang et al. [9] utilized polyvinyl alcohol and chitosan to develop a thermo-sensitive injectable hydrogel as the carrier system for treatment drugs; X. Zhao et al. [2] developed an injectable hydrogel wound dressing with conductive and self-healing properties that exhibited exceptional antibacterial activity in all experiments for wound healing; J.Y Kim et al. [10] combined gelatin hydrogel with DAA-Fe^3+^ chelate and the Fenton reaction to develop a double crosslinking gelatin hydrogel that displayed good biodegradability in simulated body fluids, showing potential for injection; M. Chen et al. [11] used aminated gelatin, adipic acid dihydrazide, and oxidized dextran to prepare an injectable and self-healing hydrogel, which combined a basic fibroblast growth factor with the disinfecting function of chlorhexidine acetate in hydrogel to accelerate cell regeneration and wound healing. Many studies have found that applying the drug carrying capacity of hydrogel to injectable systems can fill irregular surface wounds and target specific sites for treatment.

This study was inspired by mussels from the ocean, which use catechol groups and metal ions to attach to the surfaces of wet objects in fluid environments and self-heal after taking damage [12,13,14,15]. Dopa-relative polymer structures have been widely investigated in the past few decades [16,17,18,19,20,21]. Thus, development began for an injectable hydrogel that can absorb a large amount of exudate (blood, tissue fluid, etc.) for application in the healing of deep wounds. First, to address the temperature changes in the human body, the thermo-sensitive N-isopropylacrylamide (NIPAAm) monomer is used as the hydrogel substrate to control the sol-gel transition while stearyl methacrylate (C18) is added to enhance the mechanical properties. Furthermore, chelation is conducted between a dopamine acrylate with catechol group and iron ions (Fe^3+^) in a ferric chloride solution. The two are mixed and the sol-gel transition is quickly enabled through the thermo-sensitive property and action of the catechol-iron ions, which produce adhesion and self-healing effects. The study analyzes properties such as swelling, rheology, bioadhesion, and self-repair for comparison with a hydrogel that does not include catechol-iron ions.

## 2. Materials and Methods

### 2.1. Materials

The main materials used in this study are listed as follows: N,N-Diethlethanamine (Alfa Aesar, Mumbai, India); Methacryloyl chloride (Alfa Aesar, Heysham, Great Britain); Ammonium persulfate (APS, Merck, Darmstadt, Germany); N,N’-Methylenebisacrylamide (NMBA, Sigma, Burlington, MA, USA); Iron (III) chloride (Applichem, Darmstadt, Germany); N-isopropylacrylamide (NIPAAm, Fujifilm, Tokyo, Japan); N,N,N′,N′-tetramethyl-ethane-1,2-diamine (TEMED, Applichem, Darmstadt, Germany); Stearyl methacrylate (C18, Aldrich, Burlington, MA, USA); Ethyl acetate (Honeywell Burdict and Jackson, Ulsan, Korea); Sodium tetraborate (Katayama Chemical, Osaka, Japan); Boric acid (Applichem, Barcelona, Spain); Sodium dodecylsulfate (SDS, Fisher scientific, Shanghai, China); Rhodamine B (Merck, Rahway, NJ, USA); Nutrient broth (Neogen, Lansing, MI, USA); American bacteriological agar (Neogen, Lansing, MI, USA); Phosphate buffered saline (PBS, Thermo Fisher Scientific, Waltham, MA, USA). All other chemicals and reagents used were of an analytical grade.

### 2.2. Preparation of Dopamine Acrylate (DAA)

The preparation of DAA was synthesized based on the previous research [21]. We precisely weighed dopamine hydrochloride (2 g), triethylamine (1.455 mL), and methanol (20 mL) into a three-necked flask (100 mL) and stirred this for 30 min in an argon environment. We then continued stirring for 30 min in an ice bath environment (4 °C). Next, we mixed tetrahydrofuran (1.250 mL) and methacrylic acid chloride (1.110 mL) then added the solution drop by drop into the reaction mixture. Finally, we added additional triethylamine (2.080 mL) and continued stirring for four hours at room temperature. The system was continuously supplied with argon during the reaction, while being careful to avoid contact with oxygen. After the solution reacted and precipitation occurred, we then added ethyl acetate and filtered the precipitate, then added ethyl acetate and deionized water to the separatory funnel for extraction. Once extracted, we took the organic layer (upper layer) and placed it into a vacuum oven at 120 °C for one day. Finally, we obtained the dopamine acrylate powder and stored it at a temperature of −40 °C. 

### 2.3. Preparation of PNIPAAm-DAA-HA Hydrogel

We prepared a boric acid-borax buffer solution with pH 7.4 and 9.45, respectively, dissolved sodium dodecyl sulfate (SDS, 0.8 g) and sodium chloride (0.32 g) into the buffer solution (10 mL) and stirred until it was clear and transparent. Next, we dispersed stearyl methacrylate (C18, 0.1 mL) into the solution to obtain a uniform solution (10 mL, 0.01% C18). We then dissolved the monomer (NIPAAm), initiator (APS), and crosslinking agent (NMBA) into the 100 mL boric acid-borax buffer solution to complete the preparation of solution A. The molar ratio of the monomer, initiator, and cross-linking agent is 0.05:7 × 10^−4^:2.5 × 10^−4^. 

Dopamine acrylate (DAA) was then completely dissolved in 0.6 M ferric chloride solution, and the molar ratio of dopamine acrylate and ferric chloride was 1:10. This solution is called solution B. We then added solutions A and B at a ratio of 100:1 mL into a petri dish along with a small amount of promoter (TEMED) to form a PNIPAAm-DAA-HA hydrogel. Prior to gelation, both solutions are liquids with high fluidity, which thereby solves the issue of uneven distribution after gelation. Additionally, we also prepared PNIAAm hydrogel without hydrophobic groups C18 and DAA and PNIPAAm-HA hydrogel with hydrophobic groups but no DAA for comparison with the experimental group, as shown in Table 1. 

### 2.4. NMR Spectroscopy Analysis (^1^H-NMR)

We used the hydrogen spectrum to verify whether dopamine acrylate (DAA) had been successfully synthesized. We weighed 0.038 g of DAA and dissolved it in 0.5 mL DMSO-d6, then added a dropper into the NMR test tube and used a Bruker Ascend 400 MHz NMR to detect the spectrum of dopamine acrylate. 

### 2.5. Rheological Behavior of Hydrogel

We used a model PP25 rheometer (Anton Paar, MCR302, Ostfildern, Germany) to investigate the rheological behavior of the hydrogel. We then observed the impact of the crosslinking agent (NMBA) concentration on the storage modulus G′ and loss modulus G″ on PNIPAAm-HA and PNIPAAm-DAA-HA hydrogels, by using a fixed angular frequency of 10 rad/s, and varying the strain from 0.1–100% to conduct testing.

We also examined the impact of temperature on the viscoelasticity of PNIPAAm-HA and PNIPAAm-DAA-HA hydrogels, by using a fixed strain of 1%, fixed angular frequency of 10 rad/s, time of 0~900 s, and with the temperature fixed at 25 °C, 32 °C, and 37 °C to conduct testing.

### 2.6. Compression Test of Hydrogels

We prepared a cylindrical hydrogel sample with a diameter of 4.3 cm and height of 2 cm and used a microload universal testing machine (Shimadzu, AG-IS, Kyoto, Japan) to perform two types of parameter tests: using compression of 80% to test the stress that can be withstood by the three types of hydrogels: without C18 hydrophobic section, with C18 added, and with C18 and DAA added; and using a compression of 100% to test the energy that is absorbed by three types of hydrogels: without C18 hydrophobic section, with C18 added, and with C18 and DAA added. Both stress and strain were recorded.

### 2.7. Swelling Behavior of Hydrogels

We took the gelated hydrogel and used a blade to cut a piece to measure the initial weight (*W*_0_), then left and soaked the hydrogel in deionized water at room temperature. During the first three days of soaking, we used a Buchner funnel to filter the deionized water and remove the hydrogel for weighing (*W_t_*) every 12 h; after 72 h we then started weighing it every 24 h. We measured the swelling until balance was achieved or until the weight no longer increased. We calculated the swelling ratio using the following formula.
(1)$$Swelling\;ratio\;\left(\%\right)=\frac{\left(W_{t}-W_{0}\right)}{W_{0}}\times 100\%$$
*W_t_*: weight of hydrogel after swelling, *W*_0_: initial weight of hydrogel.

### 2.8. Bioadhesion Test of Hydrogels

First, we secured two pieces of pig skin on a stainless-steel plate and hung one stainless steel plate on a scale while securing another on a platform. We placed a 3 cm × 3 cm × 1 cm (length × width × thickness) piece of hydrogel in the middle of the two pieces of pig skin and attached it for 10 min. Next, was started a burette and begin dripping water, not stopping until the two pieces of pig skin were separated. Bioadhesive force—the detachment stress (kg/cm^2^)—was determined from the minimal water weight that detached two stainless steel plates [22].

### 2.9. Self-Healing Test of Hydrogels

We prepared a 6 cm × 2 cm × 5.5 mm (length × width × thickness) piece of hydrogel and stretched it with the microload universal test machine, at a tensile rate of 50 mm/min. After the hydrogel broke from stretching, we reattached the hydrogel at the broken section and left it for eight hours, 16 h, and 24 h before conducting other tensile tests. We calculated the self-healing rate (%) using the following formula.
(2)$$self-healing\;rate\;\left(\%\right)=\frac{strain\;after\;healing}{strain\;before\;healing}\times 100\%$$

### 2.10. Antibacterial Test of Hydrogels

This experiment utilized two types of gram-positive bacteria, *Bacillus cereus* (*B. cereus*) and *Staphylococcus epidermidis* (*S. epidermidis*), as well as gram-negative bacteria *Escherichia coli* (*E. coli*) in the zone of inhibition test to see whether the hydrogel prepared in this study has antibacterial properties. 

We took eight grams of nutrient broth and dissolved it in a serum bottle with one liter of deionized water and mixed it uniformly, then added 15 g of American bacteriological agar to form the test medium. Next, we placed the serum bottle in an autoclave and sterilized it for 20 min at 121 °C. We then removed the serum bottle, sprayed it with alcohol and placed the petri dish in a laminar flow hood. We poured approximately 5~10 mL of the culture medium into the petri dish and let it sit until solidification. We used a sterilized cotton swab to absorb the strain solution and spread it evenly on the surface of the solidified medium. We attached the prepared hydrogel sample onto the culture medium and placed the petri dish in an incubator to allow bacteria to grow for 24 h. We removed the petri dish and observed whether there were antibacterial properties, which was assessed by entering the results into software to measure the thickness from the sample’s edges to the inhibition zone.

### 2.11. In Vitro Transdermal Absorption Test of Hydrogels

The 2003 EU Testing and Safety Evaluation of Cosmetic Ingredients (SCCNFP/0690/03) propose using non-animal methods to reduce animal sacrifices when evaluating the safety of cosmetics [23]. Later, regulation 1223/2009 by the European Commission (EC) mandated that, starting in March 2009, local toxicity tests of cosmetic ingredients could not be performed on living organisms. Therefore, this study utilizes the in vitro transdermal absorption method to evaluate the skin’s absorption of drugs. The basic testing principles are conducted similarly to those in W. Diembeck et al. [24].

Transdermal absorption device: the device includes a supply and reception phase; the tested skin is secured above the reception phase with a clamp with the hydrogel placed on the tested skin. A heat circulating water bath maintains temperature in the system at 37 °C as shown in Figure 1.Skin preparation: this experiment uses pig skin in place of human skin for in vitro transdermal absorption testing.Drug preparation: weigh 0.479 g of Rhodamine B and dissolve in one liter of deionized water to form a 1 mM Rhodamine B aqueous solution. Next, dilute the 1 mM Rhodamine B aqueous solution to 50 μM, 10 μM, 1 μM, 0. 5 μM, 0.1 μM, and 0.01 μM for use as standard solution. A fluorescence spectrometer (Hitachi F-7000) was used to measure fluorescence to obtain the calibration line.Hydrogel preparation: soak hydrogel in a 50 μM Rhodamine B aqueous solution prepared in advance for one hour until swollen; put aside for later use.Transdermal absorption test method: cut the swollen hydrogel into round test pieces with a diameter of 1.5 cm and thickness of 3 mm. Place on a 4 cm × 4 cm (length × width) piece of pig skin and secure above the reception phase. Next, add the simulated body fluid (PBS) into the reception phase and activate the heat circulating water bath to maintain PBS at 37 °C. During the process, the reception phase must simulate blood circulation; the device is thus set up on a magnetic stirrer at a speed of 500 rpm. At fixed intervals, remove the hydrogel test piece and place into a high-pressure reactor containing 20 mL of deionized water and place the high-pressure reactor in a 90 °C oil bath to react for two hours. When the reaction is complete, let it cool to room temperature and use a fluorescence spectrometer to analyze the aqueous solution. Obtain the degree of absorption then substitute it into the calibration line to calculate concentration.

## 3. Results and Discussion

### 3.1. Formation Mechanism of PNIPAAm-DAA-HA Hydrogel

Currently, commercially available hydrogel wound dressings are primarily patches that are only suitable for shallow wounds. For deeper wounds with issues such as defects and unevenness, standard patch dressings do not offer effective healing. Most treatments require surgical sutures, skin grafts, and other treatment methods that carry risk, are time consuming, and costly. As a result, most people choose to apply topical medication to treat wounds, but this method leaves wounds open to possible issues such as infection and inflammation. Therefore, there is the need for filler that can effectively protect and treat wounds. This study combines hydrogels and injectable systems by using thermo-sensitive hydrogel and the difference between room and body temperature to perform the sol-gel transition; by developing an adhesive hydrogel with self-healing properties, it is possible to prevent the loss of wound protection from fallen or damaged dressings due to activity. Therefore, this study uses a double solution method of NIPAAm thermo-sensitive hydrogel solution and chelation Fe^3+^ ion solution that simultaneously injects both solutions to form a composite function hydrogel with adhesiveness and self-healing properties.

Studies have found that boric acid compounds with low molecular weight can be used as protective agents for glycols due to their unique electronic and physical chemistry properties; hydrogels containing boric acid have special functions such as glucose sensitivity, reversibility, and self-healing properties [25]. Additionally, the human body’s pH value is usually around 7.4; therefore, this study utilizes boric acid and borax (Na_2_B_4_O_7_·10H_2_O) to prepare a boric-acid borax buffer solution that controls the pH value of the hydrogel solution to prevent excessively high or low pH and causing secondary damage to the wound. First, a surfactant (SDS) and sodium chloride are used in a solvent to form micelles that are externally hydrophilic and internally hydrophobic. Next, we add stearyl methacrylate (C18) to form micelles, which contain C18 inside. We take the thermo-sensitive monomer (NIPAAm), initiator (APS), and crosslinking agent (NMBA) and add them to the solution with C18 micelles to form a hydrophobic association (HA) hydrogel (PNIPAAm-HA) to form solution A. Additionally, dopamine acrylate (DAA) contains a unique catechol group with *p*Ka value of 9.45 [26], the two hydroxyl groups have negative charges making them prone to attack and exhibiting strong metal chelating ability. This property is used for chelation with iron ions (Fe^3+^) in the ferric chloride solution. This is solution B, as shown in Figure 2A. We mix solutions A and B to form the PNIPAAm-DAA-HA hydrogel solution, and after adding the promoter (TEMED), NIPAAm and DAA form a binary copolymer under APS initiation, as shown in Figure 2B. This study utilizes the hydrophobic association of C18 micelles to improve the mechanical properties of the hydrogel; the chelation effect of Fe^3+^-catechol gives the hydrogel properties of adhesion and self-healing; additionally, the interaction between hydrophilic segments such as the interaction between the hydrophilic end of C18 micelles and the hydrophilic end of monomers is used; the interaction between the hydrophilic ends of DAA and the monomer; and the interaction between DAA and the hydrophilic end of C18 micelles. Through the interaction of these molecules, a stable PNIPAAm-DAA-HA hydrogel is formed, as shown in Figure 2A.

During the experiment, solution A was prone to deterioration at room temperature while solution B DAA-Fe^3+^ was prone to oxidization; thus, preparations of solutions A and B were stored at 4 °C and −40 °C, respectively. When solution A is placed in low temperature environments, the solubility of surfactant (SDS) in the solution is reduced, resulting in crystallization. Once returned to room temperature, the crystals once again dissolve and return to their initial state. For clinical application, the two solutions must be placed at room temperature so they may return to a liquid state to facilitate subsequent operations. First, inject solution B into the wound then inject solution A to mix with solution B; finally, add the promoter (TEMED) so that hydrogel can be shaped irregularly and cover the wound surface. Faced with the possibilities of external elements and human activity that could affect the wound, gel time cannot take too long, the gel must have a certain mechanical strength, and must self-heal when damaged—this paper thus explores and focuses on the study of these issues, as shown in Figure 2B.

As the promoter (TEMED) has been added into solution A, it can form into a PNIPAAm-HA hydrogel. Preliminary use of the inverted observation method found that higher concentrations of initiator (APS) resulted in faster gel time of PNIPAAm-HA. Considering that both injections require time to mix, an APS concentration of 7 × 10^−4^ M was selected to control the gel time to approximately five minutes, as shown in Table 2. Additionally, a higher NMBA concentration makes the hydrogel prone to being brittle, while when the concentration is too low, insufficient crosslinking makes it unable to maintain a fixed shape. Through experimentation, it was found that a NMBA concentration of 2.5 × 10^−4^ retains a certain degree of ductility while also being able to form varying samples based on the mold. Tests were performed based on the requirements above to determine whether mixing solutions A and B could form a PNIPAAm-DAA-HA hydrogel. First, different concentrations of FeCl_3_ were used to observe the impact of DAA-Fe^3+^ on PNIPAAm-DAA-HA hydrogel. Table 2 shows that when the FeCl_3_ concentration is at 2.5 mol, it is unable to gel; when the concentration is increased to 10 mol, it is able to gel. Based on the results of the experiment, it was found that the two solutions can gel when mixed. Next, further exploration of gel uniformity for different concentrations of the monomer (NIPAAm) was conducted. Table 2 shows that when NIPAAm concentration is greater than 0.05 mol, gel distribution is uneven; this is due to both solutions completing the sol-gel transition when they have not been mixed evenly after the promoter (TEMED) is administered. When the concentration of NIPAAm is reduced to 50 mmol, it forms an evenly distributed PNIPAAm-DAA-HA hydrogel as shown in Figure 3. It can be clearly seen that when the concentration of NIPAAm is higher than 75 mmol (Figure 3B), the gel will aggregate and cannot form a uniform whole-layer hydrogel. When the NIPAAm concentration is higher than 100 mmol, a white precipitate is found on the hydrogel (Figure 3C,D), which is due to the insufficient solubility of NIPAAm, resulting in phase separation. Based on the results of the experiment presented above, the optimized concentration of the hydrogel solution A is at NIPAAm 0.05 mol, APS 7 × 10^−4^ mol, and NMBA 2.5 × 10^−4^ mol; solution B contains DAA 1 mol and FeCl_3_ 10 mol for exploration in subsequent experiments.

### 3.2. NMR Spectroscopy (^1^H-NMR) of Dopamine Acrylate (DAA) 

This study synthesized dopamine acrylate (DAA) based on the literature [21]. In order to verify the successful synthesis in this experiment, ^1^H-NMR was used to determine changes in the structural characteristic peaks of dopamine hydrochloride and dopamine acrylate (DAA). 

Figure 4A shows the ^1^H-NMR spectroscopy of dopamine hydrochloride. δ = 2.5 ppm is the solvent peak of DMSO; δ = 3.5 is the characteristic peak of H_2_O; δ = 6.45, 6.47, 6.89 ppm is presumed to be the CH characteristic peak of the benzene ring on the catechol structure; δ = 2.71, 2.89 ppm are the two characteristic peaks of CH_2_ connecting the benzene ring and NH_2_; δ = 8.64 ppm is the characteristic peak of NH_2_; δ = 8.13 ppm is the characteristic peak of catechol structure OH. 

Figure 4B shows the ^1^H-NMR spectroscopy of dopamine acrylate (DAA). It can be observed that the structure change and chemical shift caused by the action of methacrylic acid chloride caused the original -NH_2_ to convert into -NH- δ = 7.936 ppm; δ = 5.617, 5.302 ppm is the characteristic peak of methacrylic acid chloride -CH_2_-; δ = 1.84 ppm is the characteristic peak of methacrylic acid chloride -CH_3_. Additionally, when compared with Figure 4A, it is found that the CH characteristic peaks of the benzene ring on the catechol structure are δ = 6.579, 6.619, 6.44 ppm; the two characteristic peaks of CH_2_ connecting the benzene ring and OH are δ = 2.555, 3.237 ppm; the OH characteristic peaks of catechol structures δ = 8.576, 8.640 ppm have similar signals and chemical shifts, which indicate they did not disappear during the reaction. A comparison of the chemical shifts in the above spectrum with information from the literature [21] shows a nearly identical match; it can thus be determined that this study successfully synthesized dopamine acrylate (DAA) that conformed to the expected structure.

### 3.3. Swelling and Degradation Behavior of Hydrogel

Since a significant amount of exudate seeps out of the body when wounded, PNIPAAm-DAA-HA hydrogel that will be applied to the human body must have a good degree of swelling to facilitate absorption. Therefore, we performed a swelling test by placing PNIPAAm-DAA-HA in deionized water. 

In this study, the hydrogel (PNIPAAm-HA) and the composite hydrogel of solution A and B (PNIPAAm-DAA-HA) were compared to explore the concentration of cross-linking agent (NMBA) (at concentrations of 0.25, 0.5, 0.75 mmol) and its effect on the change in swelling of both hydrogels. Figure 5A is the swelling curve of PNIPAAm-HA hydrogel. From the graph, it can be observed that the swelling degree of the hydrogel with the lowest crosslinking agent concentration after 12 h is significantly greater than that of the two other concentrations. After one day, it can be seen that higher crosslinking agent concentration results in a lower degree of swelling. At the 60th hour, 0.25 mmol concentration achieves the maximum swelling degree of 1080%; at the 48th hour, 0.5 mmol concentration achieves the maximum swelling degree of 767%; at the 36th hour, 0.75 mmol concentration achieves the maximum swelling degree of 633%. The time it takes to reach the maximum degree of swelling shows that higher concentrations of the crosslinking agent result in a lesser amount of water that can be absorbed; therefore, the reason for reaching maximum swelling faster is that high concentrations of the crosslinking agent create a hydrogel with a denser structure and a reduction in water absorption.

Figure 5B is the swelling curve of PNIPAAm-DAA-HA hydrogel. From the graph it can be observed that PNIPAAm-DAA-HA hydrogel is similar to PNIPAAm-HA hydrogel, as both exhibit a lower degree of swelling at higher concentrations of the crosslinking agent. However, PNIPAAm-DAA-HA hydrogel differs in the fact that regardless of crosslinking agent concentration, the degree of swelling almost always reaches the maximum after 36 h. The reason for this is that DAA and PNIPAAm are bonded and cross-linked in the structural arrangement so that the general structure of the hydrogels is similar in different concentrations of the crosslinking agent. However, the swelling size is still primarily affected by the crosslinking agent; 0.25 mmol concentration reaches a maximum swelling degree of 1084% at the 36th hour; 0.5 mmol reaches a maximum swelling degree of 862%; 0.75 mmol reaches a maximum swelling degree of 663%.

While hydrogel dressings exhibit excellent swelling and can absorb exudate from wounds, their degree of absorption is still limited and they must be replaced after a certain period of time. However, changing dressings for deep wounds is much more challenging compared to surface wounds. If hydrogel dressings can degrade then this can improve the issue of changing dressings for deeper wounds and prevent the possibility of bacterial infection when hydrogel is applied on wounds for prolonged periods.

From the swelling test, it can be observed that PNIPAAm-HA and PNIPAAm-DAA-HA hydrogel begin to show degradation after reaching the maximum degree of swelling. Figure 5A shows that the swelling of PNIPAAm-HA hydrogel with 0.25 mmol crosslinking agent is reduced from 1080% to 959% after 120 h; 0.5 mmol is reduced from 767% to 605%; 0.75 mmol is reduced from 633% to 388%. Figure 5B shows that the swelling of PNIPAAm-DAA-HA hydrogel with 0.25 mmol crosslinking agent is reduced from 1080% to 698%; 0.5 mmol is reduced from 862% to 555%; 0.75 mmol is reduced from 663% to 352%. The degradation is caused when hydrogel is immersed in deionized water and water molecules with smaller molecular weight and fast diffusion rates penetrate into and expand the volume of hydrogel. After a long time of swelling the hydrogel will be partially hydrolyzed, which may be partly due to the detachment of the incompletely bonded acrylate structure, thus causing a decrease in the swelling ratio of the hydrogel. From the results of the experiment presented above, it is known that the hydrogel developed in this study exhibits an excellent swelling effect as well as a degradation behavior, which solves the issue of changing wound dressings that exists in injection systems.

### 3.4. Compression Mechanical Properties and Rheological Behavior of Hydrogels

Since the hydrogel’s hydrophobic segments can slide, this disperses external stress, thereby effectively enhancing the hydrogel’s mechanical properties. In order to verify whether the addition of the hydrophobic segment (C18) improves the mechanical properties of hydrogel and disperses energy effectively, a microload universal testing machine is used to perform compression testing.

This study tested the amount of stress the three hydrogels PNIPAAm (XC), PNIPAAm-HA (CP), PNIPAAm-DAA-HA (CD) can withstand under 80% compression. The setup for the compression test is shown in Figure 6A. The stress at 80% compressive strain was collected as a histogram, as shown in Figure 6B. This shows that XC hydrogel can withstand the most stress at 1.2 × 10^−2^ N/mm^2^; CD can withstand 5.6 × 10^−3^ N/mm^2^; and CP can withstand 4.8 × 10^−3^ N/mm^2^. These results are due to the structure of XC hydrogel, which is not affected by the hydrophobic segment and displays mechanical properties that tend to be hard and brittle, resulting in more force being required to compress XC hydrogel to 80% compression. The hydrophobic section C18 in the structures of hydrogels CP and CD formed the HA micelles, which generated a slide effect, which softened the mechanical properties of both hydrogels. The CD hydrogel withstood more stress than CP, primarily as DAA and PNIPAAm are bonded and crosslinked in their structural arrangement, forming a tighter overall structure than CP, leading to a higher amount of withstood stress.

The area enclosed by the stress–strain curve represents the energy absorbed by the material under stress. Figure 6C shows that the results with regard to absorbed energy are opposite to those in Figure 6B, with CP absorbing the most energy at 9.323 × 10^−1^ N/mm^2^; next is CD at 4.957 × 10^−1^ N/mm^2^; and finally, XC at 2.763 × 10^−1^ N/mm^2^. As the mechanical properties of XC hydrogel tend towards being brittle, when the hydrogel reaches yield stress the area of curve integration is the smallest; CD is second; CP is the highest. Here, the results prove that adding the hydrophobic section C18 can effectively increase mechanical properties so that it is not as susceptible to impact and damage by external force. 

As there are a large number of non-covalent crosslinking points in the hydrogel used for the hydrophobic regions and allocation of metal ions, strong hydrophobic association can maintain the integrity of the hydrogel and stabilize ionic coordination bonds, which play a vital role in enhancing its mechanical properties. Under the action of sodium dodecyl sulfate (SDS), the stearyl methacrylate (C18) group forms micelles in the structure. Hydrophobic molecular chains can slide along micelle deformations, dissipate massive amounts of energy, and act as dynamic crosslinking points, thereby enhancing the fracture stress of hydrogel. Additionally, the interaction between iron ions and anion SDS also helps stabilize the hydrophobic association [27].

A rheometer was employed to explore the effects of PNIPAAm-HA (CP) and PNIPAAm-DAA-HA (CD) hydrogels on the viscoelasticity of colloids at different temperatures. With the hydrogel solution pH set at 7.4, the study conducted observations at 30-min intervals for the three temperatures of 25 °C, 32 °C, and 37 °C with fixed strain of 1%, and fixed angular frequency of 10 rad/s to find the changes in storage modulus (G′) and loss modulus (G″) of the hydrogel. As chelation of catechol and Fe^3+^ in the structure of DAA in PNIPAAm-DAA-HA (CD) hydrogel is strong in the basic condition, the study conducted further exploration of the impact of catechol on hydrogel when hydrogel solution pH is 9.45.

Due to the process of crosslinking, the storage modulus (G′) will continue to increase until it reaches the point of intersection with the loss modulus (G″), meaning the viscous fluid has transferred to an elastic solid. This method is used as the criterion for gel time, and the intersection point is the gel point of the hydrogel [28]. From Figure 6D, it can be observed that the storage modulus (G′) and loss modulus (G″) of CP hydrogel with pH 7.4 intersect at a point in time, and as the temperature increases there is no significant change to gel time; in Figure 6E, it is observed that the storage modulus (G′) and loss modulus (G″) of CD hydrogel with pH 7.4 intersect faster at temperature points of 32 °C and 37 °C as compared to 25 °C. It is found that CP and CD at 25 °C exhibit G′ lower than G″ at 4 min 46 s, 12 min 25 s; at 32 °C, G′ is lower than G″ at 3 min 29 s, 6 min 55 s; at 37 °C, G′ is lower than G″ at 3 min 11 s, 6 min 55 s. 

After the transition time of the hydrogel from a viscous fluid to an elastic solid the cross-linking continues, and it is necessary to determine when the hydrogel reaches complete gelation. We further use a rheometer to test the CP and CD hydrogels that have been placed for more than 30 min at a fixed angular frequency of 10 rad/s and the strain ranging from 1 to 100% to find out the complete gelation of the CP and CD hydrogels. It can be seen from Figure 6E,G that the storage modulus (G′) obtained from the strain of CP and CD hydrogels of complete gels from 1–100% reaches equilibrium at 100 Pa and 50 Pa. Comparing Figure 6D–G with each other, the complete gelation time of CP and CD hydrogels can be found. The hydrogels of CP and CD reached complete gelation at seven minutes 19 s and 14 min 22 s under the reaction at 25 °C; complete gelation was achieved at 32 °C at five minutes 51 s and eight minutes and 45 s; at 37 °C, complete gelation was reached at four minutes 40 s and eight minutes 45 s. When the hydrogel solution is injected into the wound, it needs to flow to fill the irregular wound surface, and it is considered that the gel time is within a reasonable range of about nine minutes.

For the bio-application of deep wounds, injected hydrogel dressings must flow and cover the surface of irregular shaped wounds; the acceptable range of sol-gel transition time is approximately seven minutes. When directly injected into the human body, gelling must be immediate to prevent the hydrogel solution from failing to gel on the affected area due to blood flow, and thus seven minutes is still excessive and there is much room for improvement in future applications.

### 3.5. Bioadhesion of Hydrogels 

Injectable hydrogels have gained widespread attention in recent years. As hydrogels are a material with hemostatic function, they can cover wounds while absorbing blood and exudate to protect affected areas via a barrier that prevents invasion by microorganisms while promoting wound healing. However, when faced with moist wounds and continuous bleeding, hydrogels have difficulty adhering to the tissue interface. Water molecules in blood flow can also cause hydrogen to interact with the functional groups of hydrogels and greatly weaken their adhesiveness to tissue interfaces [29]. Therefore, designing an injectable hydrogel that can maintain high adhesion even in humid environments is vital for many biomedical applications.

In order to simulate the human body, this study used readily available pig skin as the substrate to explore the adhesion of hydrogel before and after the addition of DAA-Fe^3+^ and at different concentrations of the crosslinking agent (NMBA) [22]. The whole bioadhesion experiment setup is shown in Figure 7A. Hydrogel with DAA-Fe^3+^ exhibited higher adhesiveness compared to hydrogels without. As the ratio of DAA-Fe^3+^ becomes higher, the hydrogel’s adhesiveness also increases to a greater degree. Under the conditions of the crosslinking agent being 0.25 mmol, the ratio of 1:15 DAA-Fe^3+^ is 32% higher with regard to adhesiveness than without any additives; 0.5 mmol and 0.75 mmol exhibit increases of 73% and 71%, respectively, as shown in Figure 7B. This is due to the presence of catechol groups, which allow hydrogels to exhibit higher adhesion [23]. It was also found that in addition to DAA-Fe^3+^, NMBA is also a key factor that influences adhesion strength. Hydrogels with lower concentrations of NMBA exhibit higher adhesion. When comparing with DAA-Fe^3+^, NMBA concentration at 0.25 mmol was higher than at 0.5 mmol and 0.75 mmol by 69% and 78%, respectively; DAA-Fe^3+^ at 1:10 was higher by 52%, 70%; DAA-Fe^3+^ at 1:15 was higher by 22%, 48%. This is due to the fact that low concentrations of NMBA can possess more monomers in the structure to interact with the hydrophilic segment of DAA, which in turn leads to increased adhesion. From the above results, it can be seen that the gel can exhibit good adhesion properties. In Figure 7C, the real adhesion properties of the gel are presented, including the successful adhesion to a glove, polystyrene plate and glass slide.

### 3.6. Self-Healing Properties of Hydrogels

Self-healing is a popular topic in the field of hydrogels and is closely linked to its broad application; the issue of decreased functionality due to cracks and breakage requires urgent improvement. In the past 10 years, many studies have been devoted to developing hydrogels with superior self-healing ability and good mechanical properties. This study utilizes an injectable system to apply a hydrogel dressing to deep wounds. In order to allow the application of hydrogel to any body part and adapt to daily human activity, especially in joint wounds, hydrogels with self-healing abilities will solve the problem of breaks due to frequent human activity and loss of protection.

The above adhesion test shows that the crosslinking agent (NMBA) at 0.25 mmol exhibits the best properties. The study subsequently adjusted the DAA-Fe^3+^ ratio to explore the impact of NIPAAm-DAA-HA hydrogel’s self-healing ability by using the tensile test to verify whether NIPAAm-DAA-HA hydrogel can exhibit self-healing capabilities after sustaining damage. 

Photos of self-healing tests, original gel, cut gels, re-combined gel and healing gel are shown in Figure 8A. Figure 8B is an image of the stretching test with the original and healing gels; they can stretch to a large degree of deformation. As shown in Figure 8C, for a DAA-Fe^3+^ ratio of 1:10 the maximum strain value after eight hours of repair is 1.58, a 30.6% self-healing rate compared to the initial 5.16; the condition with 1:15 exhibited a strain value of 1.5, self-healing rate of 26.5%; the condition with 1:20 exhibited a strain value of 0.93, self-healing rate of 15.81%. After 16 h, the strain value of 1:10, 1:15, and 1:20 increased to 3.89, 3.51, and 3.31 for self-healing rates of 68.37%, 57.73%, and 56.20%, respectively; after one day of repair, 1:10, 1:15, and 1:20 increased to 4.32, 5.03, 4.48 for self-healing rates of 73.3%, 75.64%, and 74.92%. All three conditions were able to exceed 70%. The experiments above confirm that the structure of broken NIPAAm-DAA-HA hydrogel can be repaired through catechol-Fe^3+^, as shown in Figure 8D,. SDS enables the diffused and randomly dispersed hydrophobic micelles in hydrogel to move closer to each other and rebuild dynamic crosslinking points, causing the surface to self-heal [21]. Thus, the hydrogel samples are able to recover a certain strength, thereby improving durability.

### 3.7. Antibacterial Properties of Hydrogels

Combining the functions of strong adhesion and antibacterial properties in hydrogels is still a challenge in moist wound conditions. While the adhesion properties of hydrogels can cover wounds, it cannot prevent the adhesion of environmental bacteria that can potentially infect wounds. Infections are a huge obstacle to the healing of wounds; severe infections can result in tissue necrosis and endanger lives. In order to accelerate the healing of wounds, the ideal dressing must possess exceptional mechanical properties of protection while also having antibacterial properties that prevent bacteria from causing secondary damage to wounds. In order to prevent the hydrogel dressing from being affected by external environments after being injected into the wound and potentially causing bacterial infection, this study focuses on bacteria commonly found on the surface of human skin and in daily life such as *Escherichia coli* (*E. coli*), *Staphylococcus epidermidis* (*S. epidermidis*), and *Bacillus cereus* (*B. cereus*) for a total of three bacterial strains. The antibacterial tests were conducted to verify whether PNIPAAm-DAA-HA hydrogel has antibacterial properties that can promote the healing of chronic wounds. 

From Figure 9A–C it is found that both PNIPAAm-HA and PNIPAAm-DAA-HA hydrogels showed zones of inhibition for *E. coli*, *B. cereus*, and *S. epidermidis*, meaning the hydrogels contain antibacterial properties. Software measurements show that PNIPAAm-HA hydrogel exhibited inhibition zone widths for *E. coli*, *B. cereus*, and *S. epidermidis* of 3.5 ± 0.08 mm, 3.9 ± 0.06 mm, and 2.8 ± 0.1 mm; PNIPAAm-DAA-HA exhibited 3.4 ± 0.08 mm, 3.5 ± 0.1 mm, and 3.1 ± 0.02 mm, as shown in the quantitative diagram.

As the results of Figure 9A–C show antibacterial properties, this study will further explore why the hydrogels are exhibiting these. The experiment is prepared with an 8 mm filter soaked in a 0.2 M boric acid aqueous solution, 0.05 M sodium tetraborate aqueous solution, pH 7.4 boric acid-borax buffer solution, and a pH 7.4 buffer aqueous solution containing hydrophobic association (HA). These four kinds of hydrogel precursors were used in zone of inhibition tests for the three bacterial strains to observe which of them contained antibacterial reactions.

The experiment results shown in Figure 9D–I indicate that the boric acid aqueous solution, borax aqueous solution, and buffer aqueous solutions did not exhibit any zones of inhibition for the three bacterial strains; the buffer aqueous solution containing HA produced a very obvious zone of inhibition. The reason for this phenomenon is that the surfactant, SDS, could perform as an antibacterial agent in a proper condition [30]. A previous report has shown that the SDS could sufficiently resist bacterial growth and showed a clear zone of inhibition during the bacteria culture experiments [31,32]. For example, they attacked the negatively charged cell walls of microorganisms to inactivate cell enzymes, causing cell lysis and death; the surface of bacteria was negatively charged and cell wall integrity was destroyed due to the neutralization reaction between the positive and negative charges. Changes in the fluidity and osmotic pressure of the cell membrane caused the death of bacteria [33].

In order to verify that hydrogel solutions with the addition of surfactants (SDS) contain antibacterial properties, PNIPAAm hydrogel without surfactants (SDS) and hydrophobic association of PNIPAAm-HA and PNIPAAm-DAA-HA hydrogels are compared through antibacterial testing. The results, shown in Figure 9J–L, show that PNIPAAm hydrogel without SDS did not exhibit zones of inhibition for the three bacterial strains of *E. coli*, *B. cereus*, and *S. epidermidis*; PNIPAAm-HA and PNIPAAm-DAA-HA hydrogels with SDS also exhibited antibacterial properties. This proves that the reason for the antibacterial properties of hydrogels prepared in this study originate from the surfactant (SDS) contained in the hydrogel solution.

### 3.8. In Vitro Transdermal Absorption and Release Behavior of Hydrogels

Compared to drugs administered through traditional oral and injection methods, transdermal drug delivery (TDD) has many advantages such as avoiding liver damage caused by drugs and sustained drug release [34]. As mentioned earlier, hydrogels are a suitable material for carrying drugs. When hydrogel dressings carry drugs, they must be able to release these to allow surface absorption by wounds to accelerate healing. For this purpose, the study used transdermal absorption to test the drug release rate of the hydrogels.

From Figure 10, it can be observed that the two hydrogels PNIPAAm-HA and PNIPAAm-DAA-HA obtain different amounts of rhodamine B release based on the length of time. PNIPAAm-HA and NIPAAm-DAA-HA hydrogels released 1.256 μg/cm^2^ and 1.639 μg/cm^2^ after two hours, with the amount released increasing with time; after 16 h, the PNIPAAm-HA and NIPAAm-DAA-HA hydrogels achieved the maximum release of approximately 6.80 μg/cm^2^ and 8.87 μg/cm^2^. According to the transdermal absorption curve in Figure 10, it can be seen that both hydrogels possess a slow-release effect, which allows hydrogel dressings to release gradually and continuously on the wound during the healing period. This will prevent issues such as a complete, one-time release of drugs, which can result in unabsorbed and lost drugs.

## 4. Conclusions

This study first uses a surfactant (SDS) and sodium chloride in a solvent to form micelles that are hydrophilic externally and hydrophobic internally. Next, stearyl methacrylate (C18) is added to form micelles, which contain C18 inside. We then take the thermo-sensitive monomer (NIPAAm), initiator (APS), and crosslinking agent (NMBA) and add them to the solution with C18 micelles to form a hydrophobic association (HA) hydrogel (PNIPAAm-HA). This is solution A. Additionally, dopamine acrylate (DAA) containing catechol functional groups is chelated with a ferric chloride (FeCl_3_) solution to form solution B. We then inject and mix solutions A and B to complete preparation of the NIPAAm-DAA-HA hydrogel. The NMR results of this study showed that when comparing synthesized DAA with pure reagent, specific characteristic peaks were detected while simultaneously maintaining the characteristic peak of pure reagent; this confirms the successful synthesis of DAA with catechol function group. The swelling test found that the presence or absence of catechol-Fe^3+^ has little effect on the maximum swelling degree of the hydrogel; the degree of swelling is primarily affected by the concentration of crosslinking agents. At the highest concentration of crosslinking agents, the swelling degree of NIPAAm-DAA-HA hydrogel can achieve 663%. The compression test pointed out that under the same stress, hydrogels containing hydrophobic group C18 micelles can absorb more energy, confirming that C18 micelles can effectively enhance the mechanical properties of hydrogels. Analysis of rheological behavior shows that gel time is shortened with increased temperature, and that at 37 °C NIPAAm-DAA-HA hydrogel can complete the sol-gel transition within seven minutes. Adhesion and self-healing tests showed that the catechol-Fe^3+^ chelation effect exhibits a certain adhesive force to pigskin and can adhere to wounds; when the hydrogel is damaged, it can self-repair under the renewed action of catechol-Fe^3+^. From the zone of inhibition and transdermal absorption tests, it is known that hydrogel possesses antibacterial properties and drug carrying functionality that can effectively accelerate the healing of wounds.

## Figures and Tables

**Figure 1 polymers-14-03346-f001:**
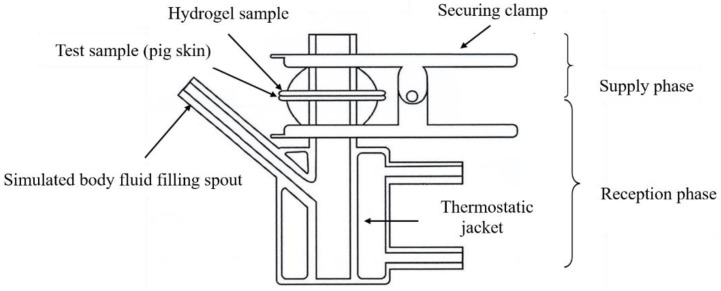
Diagram of transdermal absorption device.

**Figure 2 polymers-14-03346-f002:**
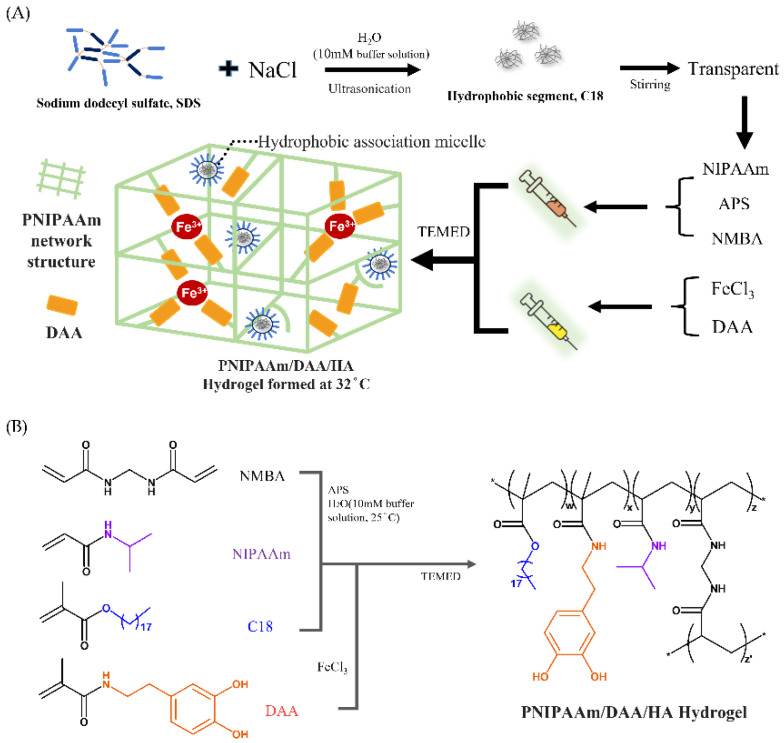
PNIPAAm-DAA-HA hydrogel, (**A**) injection formation diagram and (**B**) polymerization and chemical crosslinking structure. The synthesis of dopamine acrylamide (DAA) follows previous research [21].

**Figure 3 polymers-14-03346-f003:**
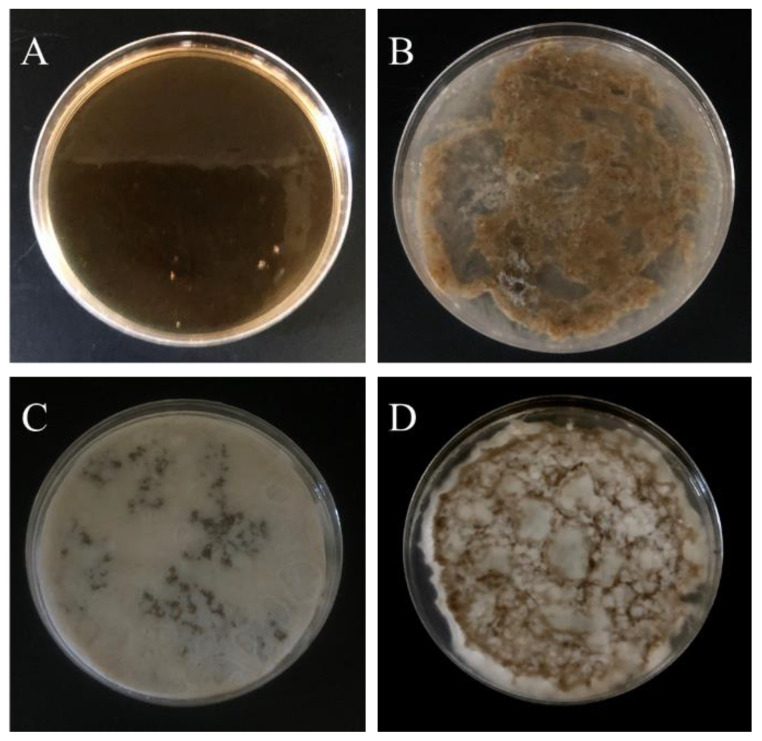
Gel results of PNIPAAm-DAA-HA hydrogel with fixed initiator (APS) 0.7 mmol, crosslinker (NMBA) 0.25 mmol with different concentrations of NIPAAm (**A**) 50 mmol, (**B**) 75 mmol, (**C**) 100 mmol, and (**D**) 200 mmol. Aggregation (**B**) and phase separation (**C**,**D**) are found in the overdose of NIPAAm.

**Figure 4 polymers-14-03346-f004:**
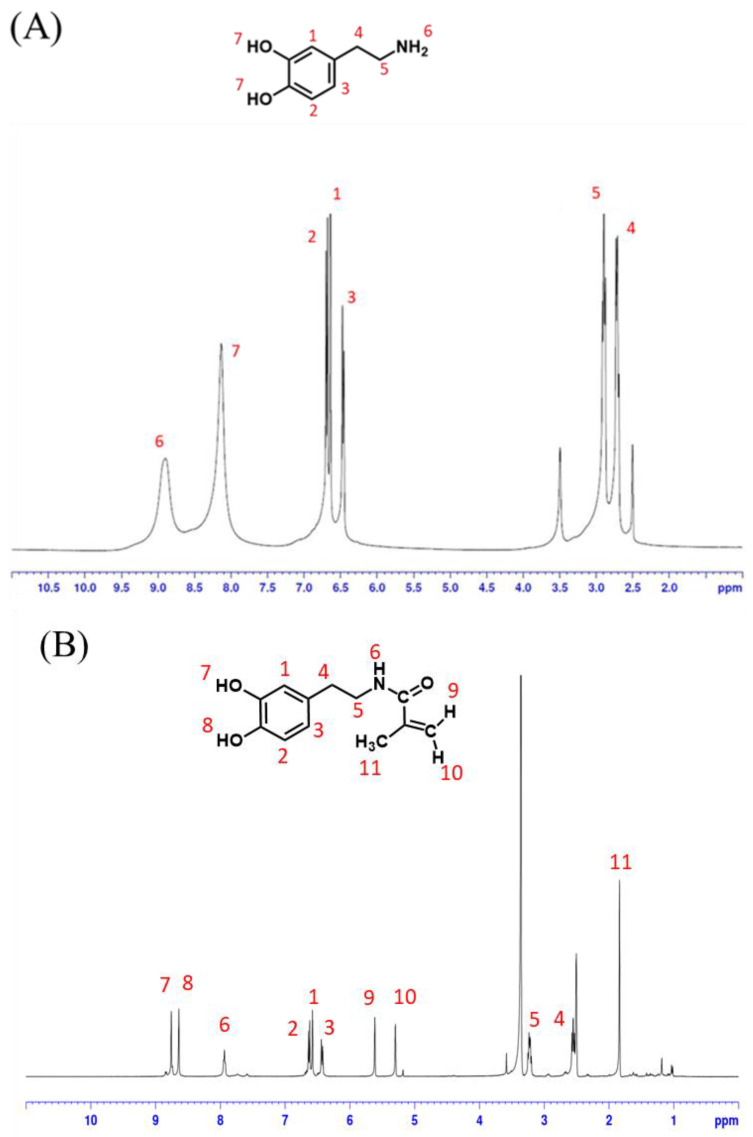
^1^H-NMR spectrum of (**A**) dopamine hydrochloride and (**B**) dopamine acrylate (DAA).

**Figure 5 polymers-14-03346-f005:**
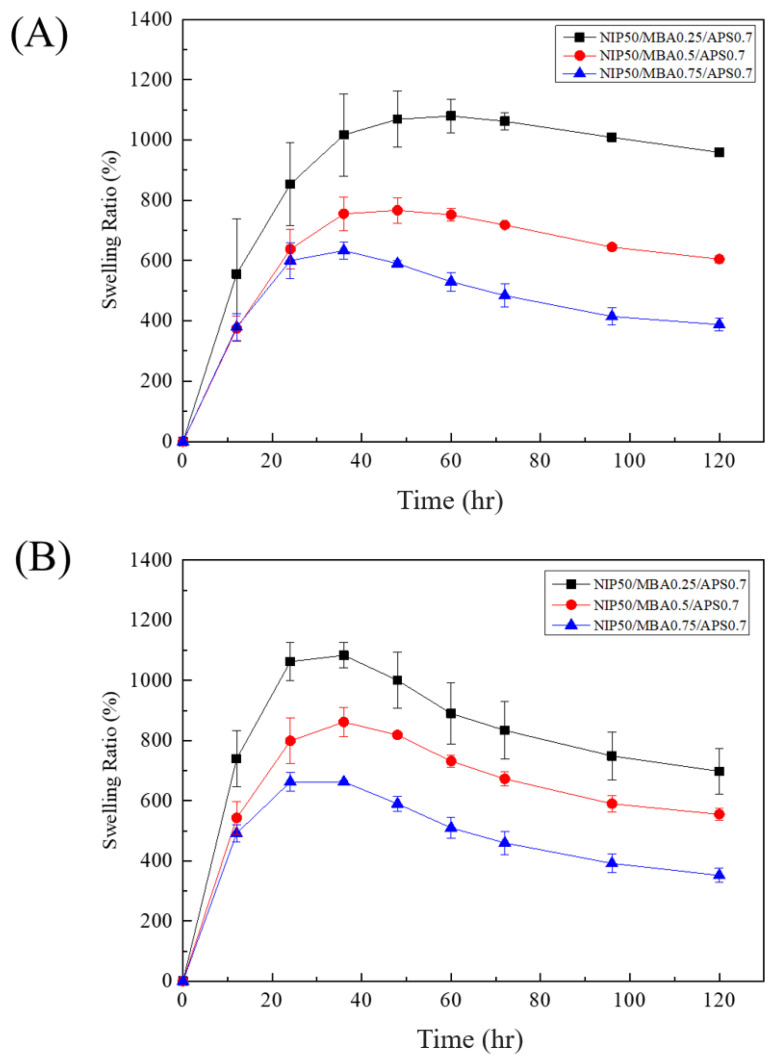
Swelling curves of (**A**) PNIPAAm-HA and (**B**) PNIPAAm-DAA-HA at different crosslinking agent concentrations of 0.25, 0.5, and 0.75 mmol.

**Figure 6 polymers-14-03346-f006:**
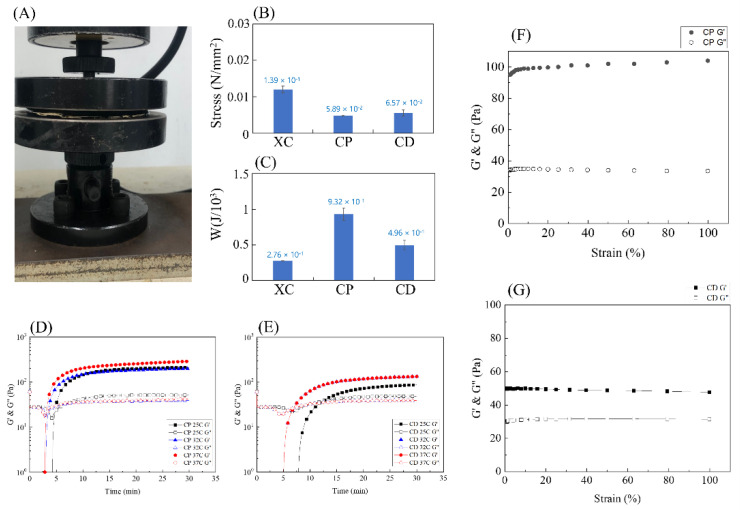
Compression and gelling points of the various hydrogels, PNIPAAm (XC), PNIPAAm-HA (CP), and PNIPAAm-DAA-HA (CD). (**A**) Compression test instrument; (**B**) withstand stress at 80% compression; (**C**) energy absorbed under compression. Evaluation of the gel points through rheological measurements: (**D**) PNIPAAm-HA (CP) and (**E**) PNIPAAm-DAA-HA (CD) hydrogels at pH 7.4. (**F**) Rheological strain curve of PNIPAAm-HA (CP), (**G**) PNIPAAm-DAA-HA (CD).

**Figure 7 polymers-14-03346-f007:**
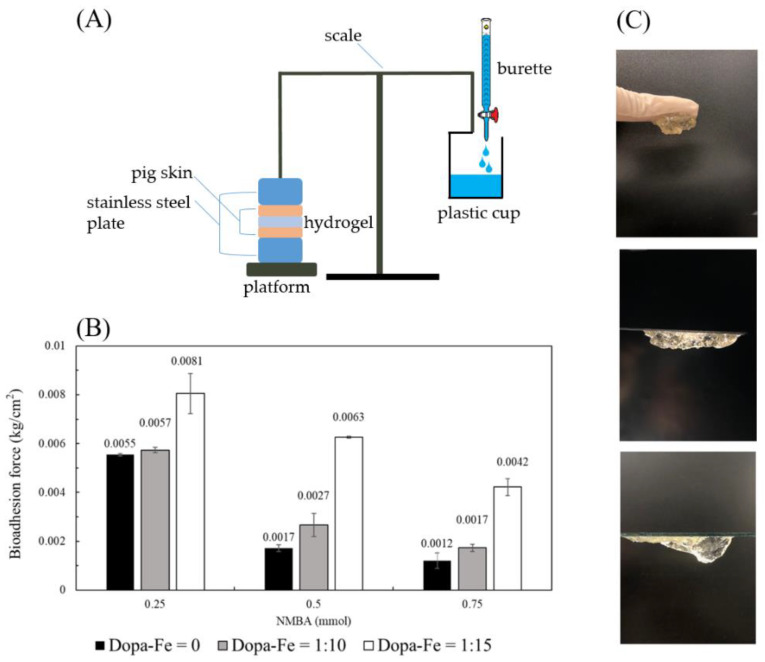
Bioadhesion testing between the hydrogel and pigskins. (**A**) Bioadhesion testing device by gravimetrical measurement. (**B**) Comparison of hydrogel bioadhesion with DAA-Fe^3+^ ratios and different concentrations of crosslinking agent (NMBA). (**C**) Image of adhesive hydrogels on the glove, polystyrene, and glass surfaces.

**Figure 8 polymers-14-03346-f008:**
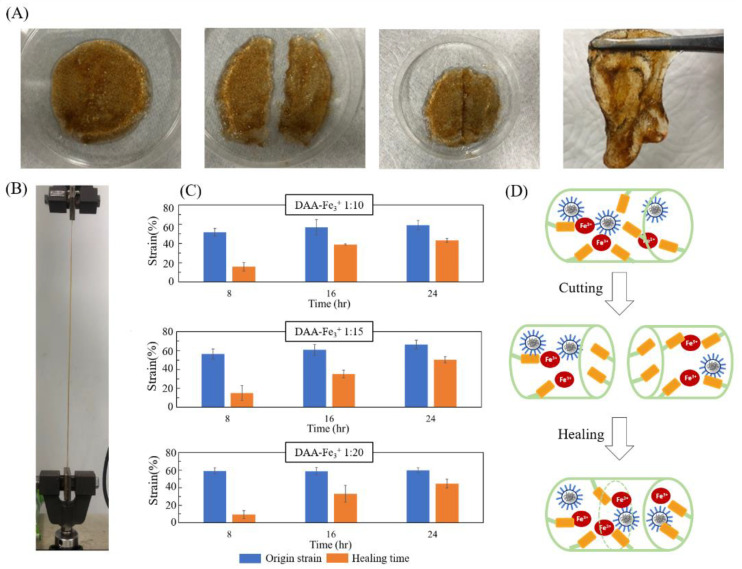
Self-healing evaluation of the NIPAAm-DAA-HA hydrogels (**A**) images of self-healing tests, original gel, cut gels, re-combined gel and healing gel (from left to right). (**B**) Image of stretching test with the origin and healing gels. (**C**) Different time strains of self-healing for NIPAAm-DAA-HA hydrogel with varied DAA-Fe^3+^ ratios, (1:10, 1:15, and 1:20). (**D**) The mechanism of self-healing for NIPAAm-DAA-HA hydrogel.

**Figure 9 polymers-14-03346-f009:**
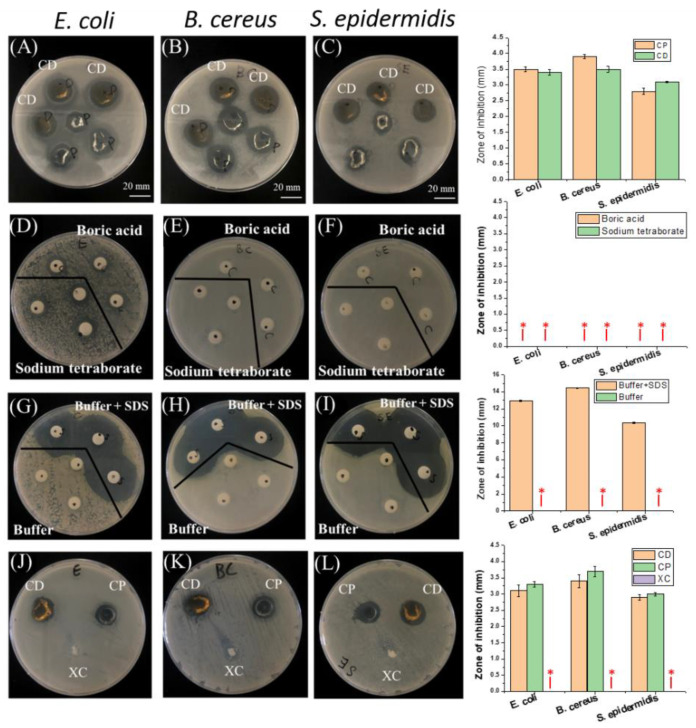
The antibacterial zone of inhibition test and quantitative characterization diagrams. PNIPAAm-HA (CP) and PNIPAAm-DAA-HA (CD) hydrogels and their antibacterial properties against (**A**) *E. coli*, (**B**) *B. cereus*, and (**C**) *S. epidermidis*. The antibacterial properties of boric acid aqueous solution and sodium tetraborate aqueous solution against (**D**) *E. coli*, (**E**) *B. cereus*, and (**F**) *S. epidermidis*. The antibacterial properties of pH 7.4 buffer solution and SDS solution with buffer against (**G**) *E. coli*, (**H**) *B. cereus*, and (**I**) *S. epidermidis*. PNIPAAm (XC), PNIPAAm-HA (CP), PNIPAAm-HA-DAA (CD) hydrogels against (**J**) *E. coli*, (**K**) *B. cereus*, and (**L**) *S. epidermidis* in terms of antibacterial properties. * Non-observation of the zone of inhibition.

**Figure 10 polymers-14-03346-f010:**
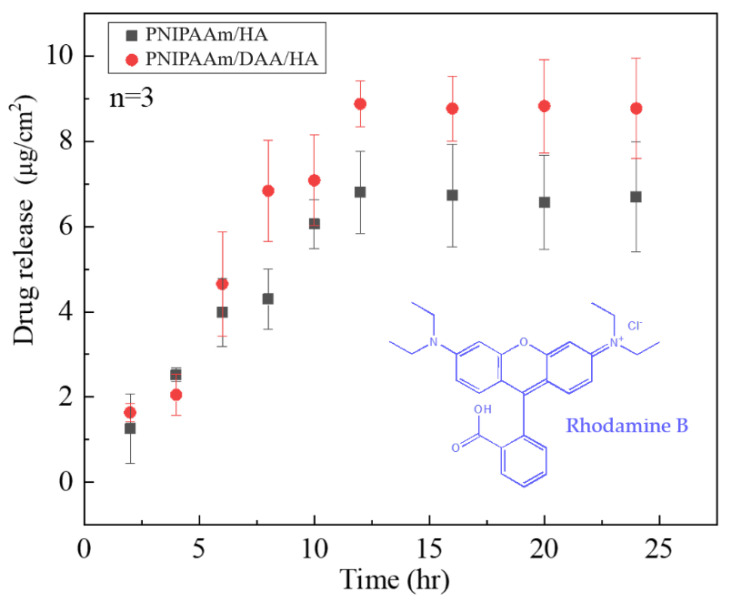
Time-dependent drug-releasing experiment of rhodamine B with controlled amount of PNIPAAm-HA (CP) and PNIPAAm-DAA-HA (CD).

**Table 1 polymers-14-03346-t001:** Formula of hydrogel samples.(“●”: with addition; “−”: without addition).

Hydrogels	Code	Hydrophobic Group C18	DAA
PNIPAAm	XC	−	−
PNIPAAm-HA	CP	●	−
PNIPAAm-DAA-HA	CD	●	●

**Table 2 polymers-14-03346-t002:** Effects of different APS concentrations on the gel time of PNIPAAm-HA; gel status of PNIPAAm-DAA-HA at different concentrations of FeCl_3_; gel status of PNIPAAm-DAA-HA at different concentrations of NIPAAm.

Conc. (mol)	Monomer (NIPAAm)	Initiatior (APS)	Crosslinking Agent (NMBA)	DAA	FeCl_3_	Gel Time (min)	Did Gelling Occur	Gel Condition
Effect of APS	0.05	3.4 × 10^4^	3	−	−	15-20	−	−
	0.05	5 × 10^4^	3	−	−	5-10	−	−
	0.05	7 × 10^4^	3	−	−	5	−	−
	0.05	1 × 10^3^	3	−	−	3	−	−
	0.05	1.5 × 10^3^	3	−	−	2	−	−
	0.05	2 × 10^3^	3	−	−	1	−	−
Effect of FeCl_3_	0.05	7 × 10^4^	2.5 × 10^4^	1	2	−	No	−
	0.05	7 × 10^4^	2.5 × 10^4^	1	5	−	No	−
	0.05	7 × 10^4^	2.5 × 10^4^	1	10	−	Yes	−
	0.05	7 × 10^4^	2.5 × 10^4^	1	15	−	Yes	−
Effect of NIPA-Am	0.05	7 × 10^4^	2.5 × 10^4^	1	10	−	−	Even
	0.075	7 × 10^4^	2.5 × 10^4^	1	10	−	−	Uneven
	0.1	7 × 10^4^	2.5 × 10^4^	1	10	−	−	Uneven
	0.2	7 × 10^4^	2.5 × 10^4^	1	10	−	−	Uneven

## Data Availability

Not applicable.

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
