# Peer review of "Mussel-Inspired Adhesive and Self-Healing Hydrogel as an Injectable Wound Dressing"

_polymers, 2022, doi:10.3390/polym14163346_

Round 1

Reviewer 1 Report

This manuscript mainly describes a multi-function hydrogel with adhesive, self-healing, antibacterial and drug release properties for injectable wound dressing.

In my opinion, this article has some problems including result discussion, logical structure and individual figures, and the corresponding explanations need to be supplemented. However, this manuscript is rich in content and has a certain reference value in the hydrogel and wound healing field. So, I suggest that this article can be published, but the paper needs major revision before acceptance for publication. My detailed comments are as follows:

  1. In this paper, the prepared hydrogel could be applied to applications in vivo, so it is necessary to supplement the biological characterization of hydrogels, such as bio compatibility, biodegradability, animal wound healing, etc.
  2. The structure of this article is vague. The dopamine acrylate prepared should be described first, then the hydrogel preparation, and finally the hydrogel properties should be charactered.
  3. The figures including Figure 3, Figure 7 and Table 2 are too small and blurry. Additionally, the text on the figures is vague and different, such as “Time (h)” and “Time (hr)”, “DAA-Fe” and “DAA-Fe3+”.
  4. In 3.8. Antibacterial properties of hydrogels, which lacks the quantitative characterization of bacteria and scale bar.
  5. In Figure 7, the horizontal axis of (A) and (C) is incomplete and the error bar is blurred.
  6. The microscopic characterization of prepared hydrogel is lacking, and it is necessary to supplement the scanning electron microscopy (SEM) characterization.
  7. The authors could consider adding the following very recent review articles into references which would again increase the interest to general mussel-inspired material (for wound healing) readers: Chemical Society Reviews‚ 2021, 50, 8319-8343; Materials Horizons‚ 2021‚8, 145-167; Chem. Soc. Rev., 2020,49, 3605-3637. Coatings, 2020, 10(7), 653. Nanoscale, 2020, 12, 1307-1324

Author Response

Suggestions:

  1. In this paper, the prepared hydrogel could be applied to applications in vivo, so it is necessary to supplement the biological characterization of hydrogels, such as bio compatibility, biodegradability, animal wound healing, etc.

Author Reply:  Thanks for referee’s suggestion. Indeed, as the reviewer said, as biological hydrogel requires more biological characterizations. The main purpose of this research is to develop a composite injectable hydrogel with adhesion and self-healing properties. In this work, a multifunctional composite hydrogel can be formed by adjusting the hydrogel solution of the AB agent. In terms of biocompatibility, we use antibacterial experiments as verification. The Nippam-DAA hybrid gel used in this study can have good antibacterial results to resist the growth of bacteria including  E. coli, B. cereus, and S. epidermidis., as shown in Figure 9. The other biological characterization, such as biocompatibility, biodegradability, and animal wound healing are the main issues that we will conduct the experiment in the coming studies.

  1. 2. The structure of this article is vague. The dopamine acrylate prepared should be described first, then the hydrogel preparation, and finally the hydrogel properties should be charactered.

Author Reply

Thanks referee for this important comment. The preparations of dopamine acrylate monomer and hydrogel have been listed in the expereimental section at page 3, line 124, section 2.2 and 2.3. This synthesis process followed the previous study [21] which has been cited in several reports. The hydrogel properties were discussed in the resulted discussion section including swelling behavior, mechanical properties and rheological behavior (section 3.3-3.4).

  1. Gao, Z.; Duan, L.; Yang, Y.; Hu, W.; Gao, G. Mussel-Inspired Tough Hydrogels with Self-Repairing and Tissue Adhesion. Appl. Surf. Sci. 2018, 427, 74–82, doi:10.1016/j.apsusc.2017.08.157.

  1. 3. The figures including Figure 3, Figure 7 and Table 2 are too small and blurry. Additionally, the text on the figures is vague and different, such as “Time (h)” and “Time (hr)”, “DAA-Fe” and “DAA-Fe3+”.

Author Reply

Thanks for this important comment and kind reminding. We modified these figures and added them into the manuscript as the following figures. Because we rearrange the figure order, the number of figures have been changed as the following presenting (Figure 3. → Figure 2.;Figure 7. → Figure 6. )  

Figure 2. PNIPAAm-DAA-HA hydrogel, (A) injection formating diagram and (B) chemical crosslinking structure. The synthesis of dopamine acrylamide (DAA) follow the previous research [21].

Figure 6.  Compression and gelling points of the various hydrogels, PNIPAAm(XC), PNIPAAm-HA(CP), and PNIPAAm-DAA-HA(CD). (A) Compression test instrument. ; (B) Withstand stress at 80% compression; (C) Energy absorbed under compression . Evaluation of the gel points through rheological measurements, (D) NIPAAm-HA (CP) and (E) NIPAAm-DAA-HA(CD) hydrogels at pH 7.4. (F) Rheological strain curve of NIPAAm-HA(CP) , (G) PNIPAAm-DAA-HA(CD).

  1. 4. In 3.8. Antibacterial properties of hydrogels, which lacks the quantitative characterization of bacteria and scale bar.

Author Reply:

Thanks for this important comment and kind reminding. Indeed, the quantitative results are important to show the differences among the conditions. We add the quantitative diagram into the antibacterial testing, as shown in Figure 9.

Figure 9. The antibacterial zone of inhibition test and quantitative characterization diagrams.  PNIPAAm-HA(CP) and PNIPAAm-DAA-HA(CD) hydrogels and their antibacterial properties against (A) E. coli, (B) B. cereus, and (C) S. epidermidis. The antibacterial properties of boric acid aqueous solution and sodium tetraborate aqueous solution against (D) E. coli, (E) B. cereus, and (F) S. epidermidis. The antibacterial properties of pH 7.4 buffer solution and SDS solution with buffer against (G) E. coli , (H) B. cereus, and (I) S. epidermidis. PNIPAAm(XC), PNIPAAm-HA(CP), PNIPAAm-HA-DAA(CD) hydrogels against (J) E. coli, (K) B. cereus, and (L) S. epidermidis in terms of antibacterial properties.

* Non-observation of the zone of inhibition

  1. 5. In Figure 7, the horizontal axis of (A) and (C) is incomplete and the error bar is blurred.

Author Reply:  Thanks for this important comment and kind reminding. We re-draw Figure 7 and combine the rheology testing to perform a better discussion for the mechanical properties of the hydrogel. We revise the axis of the figures, and also improve the resolution of the figure. Because we rearrange the figure order, the number of figures have been changed in the following presentation (Figure 7. → Figure 6. )  

  1. 6. The microscopic characterization of prepared hydrogel is lacking, and it is necessary to supplement the scanning electron microscopy (SEM) characterization.

Author ReplyThanks for this important comment and kind reminding. Indeed, in general, it is very important to observe the microstructure of hydrogel dressings by SEM. However, the multiple complex gel used in this study cannot be observed by general SEM due to its complex structure and easy absorption of water. Entering the SEM observation produces severe noise and blurring, which may be caused by the presence of iron ions. We have made a lot of efforts to find a suitable observation method. Unfortunately, limited by the equipment, we cannot get good pictures, so we can only present the hydrogel photos as shown in Figure 8.

  1. 7. The authors could consider adding the following very recent review articles into references which would again increase the interest to general mussel-inspired material (for wound healing) readers: Chemical Society Reviews‚ 2021, 50, 8319-8343; Materials Horizons‚ 2021‚8, 145-167; Chem. Soc. Rev., 2020,49, 3605-3637. Coatings, 2020, 10(7), 653. Nanoscale, 2020, 12, 1307-1324

Author ReplyThanks for this suggestion from referee. These documents are indeed important references for this study, and we have added them to the references as suggested. The references are showed below.

  1. Yang, P.; Zhu, F.; Zhang, Z.; Cheng, Y.; Wang, Z.; Li, Y. Stimuli-Responsive Polydopamine-Based Smart Materials. Chem. Soc. Rev. 2021, 50, 8319–8343, doi:10.1039/D1CS00374G.
  2. Zhang, X.; Li, Z.; Yang, P.; Duan, G.; Liu, X.; Gu, Z.; Li, Y. Polyphenol Scaffolds in Tissue Engineering. Mater. Horizons 2021, 8, 145–167, doi:10.1039/d0mh01317j.
  3. Manolakis, I.; Azhar, U. Recent Advances in Mussel-Inspired Synthetic Polymers as Marine Antifouling Coatings. Coatings 2020, 10, doi:10.3390/coatings10070653.
  4. Zhang, C.; Wu, B.; Zhou, Y.; Zhou, F.; Liu, W.; Wang, Z. Mussel-Inspired Hydrogels: From Design Principles to Promising Applications. Chem. Soc. Rev. 2020, 49, 3605–3637, doi:10.1039/c9cs00849g.
  5. Guo, Q.; Chen, J.; Wang, J.; Zeng, H.; Yu, J. Recent Progress in Synthesis and Application of Mussel-Inspired Adhesives. Nanoscale 2020, 12, 1307–1324, doi:10.1039/c9nr09780e.

Reviewer 2 Report

The present manuscript describes the preparation and study of a self-healing hydrogel as material for injectable wound dressing.

The design of the work, as well as some results, are of interest. However, the manuscript suffers from serious drawbacks:

  1. Though I am not a native English-speaker, I find that the use of English language is very bad and the manuscript should be rewritten in many cases.
  2. In addition to the previous remark, almost the whole experimental section, as well as a significant part of the Results and Discussion and Conclusion sections are written as if they are addressed to technicians and not to the relevant scientific community.
  3. Figure 3B is not correct. In a free radical polymerization process all monomers (NIPAAm, NMBA, DA and C18) are simultaneously copolymerized. In Figure 3B, copolymerization is a step process, while C18 is not copolymerized though it is a methacrylate monomer.
  4. The swell study was performed at room temperature? What happens at the body temperature?
  5. Swelling study. The authors find a decrease of swelling for large times and discuss this point in terms of degradation, arguing that “water molecules will attack” polymers. I can hardly understand this argument, considering that this is a synthetic hydrogel based on (meth)acrylate structural units.
  6. Section 3.8. In lines 576-578, the authors claim that “The reason for this phenomenon is that the surface active agent (SDS) and sodium chloride added before the formation of HA makes the solution carry positively charged ions”. I can not understand this argument, since SDS is a negatively charged surfactant.
  7. As far as I understand, for the application of this system, solution A will be in direct contact with the human body and no purification could be performed from residual monomers. If this is correct, I wonder whether this could be acceptable, since solution A contains (meth)acrylic monomers, notably NIPAAm and NMBA.

In view of the aforementioned remarks, I recommend major revision. However, my opinion is that the manuscript should be rewritten, in order to meet the quality requirements, both as it concerns presentation/discussion and, in some cases, scientific arguments. 

Author Response

Detailed comments are as follows:
1. Though I am not a native English-speaker, I find that the use of English language is very bad and the manuscript should be rewritten in many cases.

Author ReplyThanks for referee’s reminding. We rechecked all paragraphs, and reworded sentences to improve readability. And redraw and add descriptions to enhance the richness and concatenation of the paper. Please referee again to reconsider the English writing of this article.

  1. 2. In addition to the previous remark, almost the whole experimental section, as well as a significant part of the Results and Discussion and Conclusion sections are written as if they are addressed to technicians and not to the relevant scientific community.

Author ReplyThanks for referee’s reminding. We have added additional descriptions and discussions to each paragraphs throughout the article. We also try to add descriptions to connect our relevant arguments/main ideas, including the following points.

  1. This technology is an injectable gel technology. The thermoresponsive gel solution formed by agent A and agent B can quickly be gelling to form the network structure.
  2. The hydrophobic association (HA) micelle formed by adding SDS and C18 structure can increase the gel strength through hydrophobic interaction.
  3. The dopamine-Fe3+ structure of DAA can increase the self-healing property and bioadhesion ability of the gel.
  4. The composite hybrid hydrogel can have the good antibacterial ability and drug release ability, and has a wide range of applications.

  1. 3. Figure 3B is not correct. In a free radical polymerization process, all monomers (NIPAAm, NMBA, DA and C18) are simultaneously copolymerized. In Figure 3B, copolymerization is a step process, while C18 is not copolymerized though it is a methacrylate monomer.

Author Reply:   Thanks for referee’s reminding. We are sorry that we do not have a clear expression of the reaction mechanism, we have proposed a revision version, and we have re-drawn the reaction mechanism diagram of the gel, as shown in Figure 2. SDS will first form a hydrophobic association micelle with the C18, and then be mixed into agent A. Therefore, in this free-radical crosslink-gelling polymerization, the reacted monomers are NIPAAm, NMBA, and DAA. C18 is not an acrylate-based monomer and does not participate in the formation of the complex hydrogel structure.

Figure 2. PNIPAAm-DAA-HA hydrogel, (A) injection formating diagram and (B) chemical crosslinking structure. The synthesis of dopamine acrylamide (DAA) follow the previous research [21].

  1. 4. The swell study was performed at room temperature? What happens at the body temperature?

Author ReplyThanks for referee’s reminding. Indeed, for implantable medical materials or invasive gels, the swelling behavior for body temperature at 37˚C is very important. In this study, however, we used an injectable rapid-gel technique with the goal of rapidly gelling the mixed reaction solution on the skin surface. Previous studies have shown that the temperature of the skin surface is about 30˚C~32˚C. This temperature is exactly the room temperature of the research team's laboratory. Therefore, the swelling testing of this study is to simulate the room temperature of the skin surface as the temperature control.

  1. 5. Swelling study. The authors find a decrease of swelling for large times and discuss this point in terms of degradation, arguing that “water molecules will attack” polymers. I can hardly understand this argument, considering that this is a synthetic hydrogel based on (meth)acrylate structural units.

Author Reply:  Thanks for referee’s reminding. We apologize for the incomplete description in this part causes misunderstanding. After a long time of swelling, the hydrogel will be partially hydrolyzed, which may be partly due to the detachment of the incompletely bonded acrylate structure, thus causing the decrease of swelling ratio of the hydrogel. We have revised the description of this section as follows:

At page 11, line 34,

“Through a long time of swelling, the hydrogel will be partially hydrolyzed, which may be partly due to the detachment of the incompletely bonded acrylate structure, thus causing the decrease of swelling ratio of the hydrogel. From the experiment results above, it is known that the hydrogel developed in this study exhibits an excellent swelling effect as well as a degradation behavior which solves the issue of changing wound dressings that exists in injection systems.”

  1. 6. Section 3.8. In lines 576-578, the authors claim that “The reason for this phenomenon is that the surface active agent (SDS) and sodium chloride added before the formation of HA makes the solution carry positively charged ions”. I can not understand this argument, since SDS is a negatively charged surfactant.

Author ReplyThanks for referee’s reminding.  In this part of the discussion, we have a wrong description, so we re-correct this paragraph. In many previous studies, biosurfactants have been mentioned to have good antibacterial properties. As described in the report of Mayri etc., the buffer solution prepared by SDS can inhibit the effects of gram-positive and gram-negative. We revised the description in the paragraph and cited the related references, as shown below.

At page 17, line 22,

“From the experiment results of Figure 9 (D)(E)(F) and (G)(H)(I) showed that the boric acid aqueous solution, borax aqueous solution, and buffer aqueous solutions did not exhibit any zone of inhibitions for the 3 bacterial strains; the buffer aqueous solution containing HA produced a very obvious zone of inhibition. The reason for this phenomenon is that the biosurfactant, SDS, could perform as an antibacterial agent in a proper condition. Previous report has shown that the biosurfactants sufficiently resist bacterial growth and shown a clear zone of inhibition during the bacteria culture experiments [31–33].”

  1. Omurzak, E.; Tegin, R.A.A.; Kyzy, A.B.; Satyvaldiev, A.; Zhasnakunov, Z.; Umetova, G.; Kelgenbaeva, Z.; Abdullaeva, Z.; Mashimo, T. Effect of Surfactant Materials to Nanoparticles Formation under Pulsed Plasma Conditions and Their Antibacterial Properties. Mater. Today Proc. 2018, 5, 15686–15695, doi:10.1016/j.matpr.2018.04.179.
  2. Harshada, K. Biosurfactant: A Potent Antimicrobial Agent. J. Microbiol. Exp. 2014, 1, 173–177, doi:10.15406/jmen.2014.01.00031.
  3. Díaz De Rienzo, M.A.; Stevenson, P.; Marchant, R.; Banat, I.M. Antibacterial Properties of Biosurfactants against Selected Gram-Positive and -Negative Bacteria. FEMS Microbiol. Lett. 2016, 363, fnv224, doi:10.1093/femsle/fnv224.

  1. 7. As far as I understand, for the application of this system, solution A will be in direct contact with the human body and no purification could be performed from residual monomers. If this is correct, I wonder whether this could be acceptable, since solution A contains (meth)acrylic monomers, notably NIPAAm and NMBA.

Author ReplyThanks for referee’s reminding. Indeed, the biocompatibility is an important problem for biomedical application.  In this study, by controlling the reaction ratio and speed, the gel formation can be completed quickly, so that the unreacted substances can reach a state that will not easily release to harm the human body. The main goal of this study is to design an injectable hydrogel structure that can be rapidly coagulated by temperature responsive ability. However, the biocompatibility experiments of this study are limited by experimental limitations and cannot be fully performed. It may be possible to add biocompatible monomer structures, such as PEG or zwitterionic structures in the extend researches in the future.

------------------------------------------------

We hope that our answers and modifications will make our intentions and manuscript clearer. If editor and reviewers have any supplementary questions, comments or concerns, please do not hesitate to let us know.

Thank you very much.

Sincerely,

Ying-Nien Chou

Department of Chemical and Materials Engineering, Southern Taiwan University of Science and Technology, Tainan 71005, Taiwan

Round 2

Reviewer 2 Report

Please, see attached file

Author Response

The authors have made some efforts to improve the manuscript, following the remarks. However, in

many crucial cases either they misunderstood, or they did not understand the remark.

For example (enumeration of the remarks according to previous review):

Detailed comments are as follows:
Remark 2. This remark did not refer to add more details. Just, not to use so much imperative form, especially in the experimental section.

Q in 1st reviewing : [In addition to the previous remark, almost the whole experimental section, as well as a significant part of the Results and Discussion and Conclusion sections are written as if they are addressed to technicians and not to the relevant scientific community.]

Author ReplyThanks for referee’s comment and explanation.  We send this article for professional English editing by a native speaker. Especially, we also ask the editor to focus on the grammatical corrections for the experimental section. In this revision, we send the proofreading version and the final collection version of manuscript to show the English editing. More detailed revisions please refer to “2. Material and methods” in the new version of manuscript. The attached file includes the following parts.

Final collection manuscript

‚Proofreading version

ƒCertificate of the editing “Paul Steed Certification of English Proofreading “

Remark 3. The chemistry shown in Figure 2B is unacceptable for any polymer chemist and it can not appear in a journal such as polymers. Free radical copolymerization is not a stepwise process. All monomers are reacting simultaneously. Still, the authors show this reaction in two steps.  Moreover, I can not understand the argument concerning C18.  As stated in section 2.1, line 117 C18 is a methacrylate monomer and a, in principle, it can be copolymerized.

Q in 1st reviewing : [Figure 3B is not correct. In a free radical polymerization process, all monomers (NIPAAm, NMBA, DA and C18) are simultaneously copolymerized. In Figure 3B, copolymerization is a step process, while C18 is not copolymerized though it is a methacrylate monomer.]

Author ReplyThanks for referee’s reminding.  The original presentation of the flow chart was intended to express the effects of agent A and agent B, so it was divided into multi-segment to perform the reaction. However, as the reviewer said, the performing way will confuse the reader, since we are in fact processing a one-step free radical polymerization after mixing. Therefore, we revised Figure 2B according to the reviewer's suggestion to present free radical polymerization and crosslinking in a simultaneous reaction.

For the part of C18, the main purpose of C18 is to form HA micelles with SDS and provide hydrophobic associate interaction to increase gel strength when forming hydrogel. Indeed, as the reviewer stated, we mistakenly neglected the polymerization linkage provided by the acrylate structure. Thanks to reviewer for pointing out our mistake, we revised our structure and added C18 to the main polymer chain structure of the copolymer, as shown in the revised Figure 2B.

Figure 2. PNIPAAm-DAA-HA hydrogel, (A) injection formation diagram and (B) polymerization and chemical crosslinking structure. The synthesis of dopamine acrylamide (DAA) follows previous research [21].

Remark 6. The authors tried to make appropriate changes. However, I do not understand many things in their effort to give an explanation. First, unless I missed it, no reference to a work in Mayri etc. is cited.  Second, why SDS is called a biosurfactant? Is there any reference that pure SDS is an effective antibacterial agent? In fact, in ref. 32 the term “SDS” is not used, while in ref. 31 it is shown that SDS has no suppression zone (Figure 10).

Q in 1st reviewing : [Section 3.8. In lines 576-578, the authors claim that “The reason for this phenomenon is that the surface active agent (SDS) and sodium chloride added before the formation of HA makes the solution carry positively charged ions”. I can not understand this argument, since SDS is a negatively charged surfactant.]

Author Reply:   Thanks for referee’s reminding.  We apologize for the inconvenience caused to reviewers (reader). First of all, Mayri A. D´ıaz De Rienzo is the first author of the original ref. 33 (Antibacterial Properties of Biosurfactants against Selected Gram-Positive and -Negative Bacteria), as the listed picture below. This is a problem with the order and abbreviations when performing citations, resulting in no "Mayri" in the last citation, so we revise the author abbreviations used in the paper. They perform the antibacterial properties and ability to disrupt biofilms of various surfactants including sodium dodecyl sulphate (SDS). The article shows that SDS could against the formation of biofilm.

According to the reviewer's question, we removed the word "biosurfactant" to avoid misleading readers.  For a more efficient explanation, we have rewritten this paragraph and re-cited the new literature, focusing on the antimicrobial behavior of SDS and related citations.  Surfactants with antibacterial effects have been extensively studied. Among the various antimicrobial surfactants, anionic surfactant also has antibacterial properties [31]. The way it destroys the cell membrane is from the incorporation of detergent. Many previous studies have pointed out that SDS has good antibacterial properties, because SDS has the ability to destroy the cell membrane of bacteria, and is often selected as a reagent for destroying cells in biological experiments. You can also refer to this video about the antibacterial properties of SDS. httpswww.youtube.comwatchv=Vi-2IcXS9mk

The previous report published by Li et al. was pointed out that when SDS exists in a porous polymer, it can effectively resist the formation of E. coli. biofilm [32]. In the study of Díaz et al., it was found that SDS can produce a selective zone of inhibition for B. subtilis.[33].

At page 17, line 22,

“From the experiment results of Figure 9 (D)(E)(F) and (G)(H)(I) showed that the boric acid aqueous solution, borax aqueous solution, and buffer aqueous solutions did not exhibit any zone of inhibitions for the 3 bacterial strains; the buffer aqueous solution containing HA produced a very obvious zone of inhibition. The reason for this phenomenon is that the surfactant, SDS, could perform as an antibacterial agent in a proper condition [31]. Previous report has shown that the SDS could sufficiently resist bacterial growth and shown a clear zone of inhibition during the bacteria culture experiments [32,33].”

  1. Omurzak, E.; Tegin, R.A.A.; Kyzy, A.B.; Satyvaldiev, A.; Zhasnakunov, Z.; Umetova, G.; Kelgenbaeva, Z.; Abdullaeva, Z.; Mashimo, T. Effect of Surfactant Materials to Nanoparticles Formation under Pulsed Plasma Conditions and Their Antibacterial Properties. Mater. Today Proc. 2018, 5, 15686–15695, doi:10.1016/j.matpr.2018.04.179.
  2. Harshada, K. Biosurfactant: A Potent Antimicrobial Agent. J. Microbiol. Exp. 2014, 1, 173–177, doi:10.15406/jmen.2014.01.00031.
  3. Díaz De Rienzo, M.A.; Stevenson, P.; Marchant, R.; Banat, I.M. Antibacterial Properties of Biosurfactants against Selected Gram-Positive and -Negative Bacteria. FEMS Microbiol. Lett. 2016, 363, fnv224, doi:10.1093/femsle/fnv224.

Remark 7. Though I do not find convincing the initial argument (“..unreacted substances can reach a state that will not easily release..”), I accept that the present work is a proof of concept and the biocompatibility aspects should be ameliorated. 

Q in 1st reviewing : [As far as I understand, for the application of this system, solution A will be in direct contact with the human body and no purification could be performed from residual monomers. If this is correct, I wonder whether this could be acceptable, since solution A contains (meth)acrylic monomers, notably NIPAAm and NMBA.]

Author ReplyThanks for referee’s reminding.  As mentioned earlier, the main technical and scientific contributions of this study come from the following points: 1). Injectable hydrophobic association gels 2). Thermally sensitive and rapid gel formation 3). Adhesion and self-healing properties due to biomimetic structures 4). Capable used as a drug-releasing gel. In this study, a composite gel with multiple properties was obtained through the control of parameters. However, due to the limitation of experimental conditions, biocompatibility experiments have not been able to be increased. In this study, more biocompatible chemical structures will be combined based on existing technologies in the future to enhance practical applicability.

------------------------------------------------

We hope that our answers and modifications will make our intentions and manuscript clearer. If editor and reviewers have any supplementary questions, comments or concerns, please do not hesitate to let us know.

Thank you very much.

Sincerely,

Ying-Nien Chou, Hong-Ru Lin

Department of Chemical and Materials Engineering, Southern Taiwan University of Science and Technology, Tainan 71005, Taiwan
